# Is CT Still the Gold Standard in Semicircular Canal Dehiscence? Diagnostic Value of MRI in Poschl and Stenver Planes

**DOI:** 10.3390/brainsci15060555

**Published:** 2025-05-23

**Authors:** Cagatay Bolgen, Birsen Unal Daphan

**Affiliations:** 1Interventional Radiology Department, Adana Medline Hospital, 01170 Adana, Turkey; 2Radiology Department, School of Medicine, Kirikkale University, 71450 Kirikkale, Turkey; drbirsenunal@gmail.com

**Keywords:** magnetic resonance, computed tomography, Stenver, Poschl, semicircular canal dehiscence, posterior semicircular canal, superior semicircular canal, comparison

## Abstract

Background/Objectives: The primary aim of this study was to investigate whether magnetic resonance imaging (MRI) of the superior and posterior semicircular canals (SCs) in cases with and without dehiscence gives results similar to those of CT. As a novel contribution, the secondary aim was to assess the diagnostic correlation between CT and MRI sequences obtained primarily in Poschl and Stenver planes, instead of reformatted images, for detecting superior and posterior semicircular canal dehiscence. Methods: A total of 103 patients were retrospectively evaluated based on CT scans, and 27 of them, with the appearance or suspicion of at least one SCD and/or thinner-than-normal canal roof bone, were prospectively examined with MRI. Results: With CT as a reference, MRI had a 78% detection rate and 92% specificity for the detection of dehiscence in the superior SCs. For posterior SCs, the dehiscence detection rate and specificity of MRI were 70% and 97%, respectively. CT and MRI examinations showed a significant agreement in the diagnosis of SCD (κ = 0.71, *p* < 0.001 for superior SCD; κ = 0.73, *p* < 0.001 for posterior SCD). The agreement values of MRIs obtained in Poschl and Stenver planes with CT in the detection of dehiscence were calculated as κ = 0.43 in Poschl and κ = 0.51 in Stenver for superior SCD; κ = 0.45 in Poschl and κ = 0.46 in Stenver for posterior SCD. Conclusions: The MRI results demonstrated similar diagnostic precision to CT when identifying SCD. In patients presenting with vertigo, nystagmus, and hearing loss, normal MRI findings may be sufficient to exclude semicircular canal dehiscence (SCD), thereby potentially obviating the need for additional CT imaging. The newly introduced Poschl and Stenver plane MRI sequences demonstrate a moderate relationship with CT for SCD diagnosis.

## 1. Introduction

Semicircular canal dehiscence (SCD) is a clinical condition resulting from bone defects in the otic capsule of the inner ear. It was first defined as superior semicircular canal dehiscence (SSCD), and then started to be evaluated in a spectrum including posterior and lateral canals. SCD causes abnormal hydroacoustic conduction pathways across the cochlea and labyrinth, forming the ‘third window’ of the labyrinth [1]. The described defect leads to findings such as a low-frequency air–bone gap and nystagmus due to sound/pressure [2]. Patients complain of vertigo, balance disorders, hyperacusis and hearing loss [1]. SSCD is seen in 0.5% of cadaveric specimens, while posterior semicircular canal dehiscence (PSCD) is seen in 2% of adults and 1.3–43% of children [2].

High-resolution computed tomography (CT) is the gold standard in imaging diagnosis [3], and tests such as video head impulse test (vHIT) are also used [2]. A definitive diagnosis is made by surgical intervention and histopathology. Before deciding on surgery, dehiscence should be seen radiologically and this preliminary diagnosis should be strengthened with some audiological and neurophysiological tests (such as vestibular evoked myogenic potentials (VEMPs)) [4]. Middle fossa craniotomy and transmastoid approaches are used in the treatment, and the middle fossa approach has been shown to be more effective [5].

Although CT is the gold standard in the imaging diagnosis of SCD, it may yield false positive results—reported in up to 27% of cases—particularly when the bony covering of the superior semicircular canal is thin or due to volume-averaging artifacts, leading to potential overdiagnosis, and it also involves exposure to ionizing radiation [1,6]. Magnetic resonance imaging (MRI) may be an alternative with its soft tissue contrast and radiation-free advantages, but its role in the diagnosis of SCD has not been sufficiently investigated. There is a lack of information in the literature about the effectiveness of MRI sequences, especially in Poschl and Stenver planes. The lack of standardized guidelines for SCD leads to different diagnostic and therapeutic approaches. There is also a lack of information about multiple otic capsule dehiscences, and the role of MRI in the detection of multiple dehiscences has not been investigated [7]. Although Lee et al. [8] developed a three-dimensional MRI method to examine inner ear changes after surgery, a systematic evaluation of MRI-CT correlation is lacking.

The primary aim of this study was to evaluate whether MRI gives results close to CT by performing MRI on semicircular canals (SCs) with and without dehiscence identified on CT. When the studies investigating the diagnostic adequacy of MRI in SCD are examined in the literature, it is seen that the Poschl and Stenver plan sequences are not obtained by planning during the examination but by the reconstruction method over axial and coronal sequences in the post-processing stage. The secondary aim of our study was to investigate the correlation between CT findings and MRI images obtained directly in the Poschl and Stenver oblique coronal planes, which were acquired as part of the primary MRI protocol. This approach differs from previous studies that relied on post-processing reformatted images and allows for a more accurate assessment of semicircular canal dehiscence.

## 2. Materials and Methods

### 2.1. Study Population and Sample

This study was designed as a retrospective and prospective cross-sectional study. A total of 103 consecutive patients who underwent computed tomography (CT) examination of the temporal region in the Department of Radiology between March and August 2012 were retrospectively evaluated. The radiological evaluation of the CT scans was conducted by an experienced radiologist to identify patients with potential semicircular canal (SC) dehiscence. The patients were included in the prospective part of this study if they had at least one SC with either (i) a clear bony defect (definite dehiscence), (ii) suspicious findings suggestive of dehiscence (suspected dehiscence), or (iii) a markedly thinned bony roof (≤0.5 mm) without visible discontinuity (thin bone roof). Patients whose CT scans showed completely normal bony coverage of all semicircular canal roofs were excluded. These imaging-based criteria served as the sole basis for inclusion. No additional restrictions were applied regarding age, sex, or other demographic or clinical characteristics. To reduce potential selection bias, all eligible patients were consecutively included in the MRI evaluation. Magnetic resonance imaging (MRI) was prospectively performed in the 27 eligible patients between September and October 2012. In one patient, the right superior semicircular canal (SSC) could not be adequately visualized due to insufficient image quality; therefore, only this canal was excluded from the analysis, while the remaining canals of the patient were included. Consequently, a total of 53 superior and 54 posterior semicircular canals were evaluated.

The classification used in this study for radiological assessment of SSC dehiscence was adapted from previously published definitions and standardized for consistency. Accordingly: (i) Definite dehiscence refers to a focal discontinuity in the bony roof of the SSC clearly visible on both CT and MRI. (ii) Suspected dehiscence refers to an indistinct defect suggestive of discontinuity but not confirmed in two planes. (iii) Thin bone roof refers to an intact but attenuated structure with a thickness of 0.5 mm or less, without any overt defect. This classification system was used to enhance diagnostic accuracy and strengthen the clinical–radiological correlation [9,10].

### 2.2. Study Procedures

Within the scope of this study, MRI was prospectively performed in patients with presence or suspicion of SCD detected by CT. In addition to standard axial and coronal images, MRI scans were performed including oblique coronal sections in Poschl and Stenver plans, which were planned and obtained during the scan for the first time in the literature. The CT scan results of all patients were evaluated in detail with images in Poschl and Stenver oblique planes in addition to conventional MR plans obtained in axial and coronal planes.

The evaluation of the images was performed blinded by two independent radiologists using standardized criteria. Interobserver agreement was assessed using Cohen’s kappa coefficient. A kappa value between 0.00 and 0.20 was interpreted as poor agreement, 0.21–0.40 as moderate agreement, 0.41–0.60 as good agreement, 0.61–0.80 as very good agreement, and 0.81–1.00 as excellent agreement. The validity and reliability of the CT and MRI imaging methods used in the data collection process have been confirmed in previous studies in the literature [11]. Temporal CT examinations were obtained on a 2-detector CT device (General Electric, Hispeed Dual, Chicago, IL, USA) with a slice thickness of 1 mm.

MRI examinations were performed with a 1.5 Tesla MRI device (Intera Master, Philips Medical Systems, Cleveland, OH, USA) and images were acquired with software version R5.1V1L1 (Philips Medical Systems, Best, Netherlands) on View Forum workstation with the following parameters: T2A 3D-TSE in axial, coronal, Poschl (Figure 1) and Stenver (Figure 2) planes (TR/TE: 1500 ms/250 ms, imaging matrix: 240 × 256, slice thickness: 1.0 mm, slice spacing: 0.5 mm, turbo factor: 74, FOV: 180 mm, voxel volume: 0.75 × 0.94 × 1.0 mm, NEX 2, total scan time: 335 s, number of slices: 60, FA: 90°).

The data of this study were retrieved from the Specialization Thesis of the first author [12].

### 2.3. Intervention Protocol or Study Groups

Our research did not involve any surgical procedures or invasive medical interventions. MRI functions as a diagnostic tool for radiology without causing any invasion to the patient’s body. This research study divided the participants into two main groups based on their CT scan findings: Group 1 included patients with semicircular canal dehiscence (SCD) or suspected SCD, while Group 2 included patients without SCD. This grouping was determined by the evaluation of high-resolution CT images by two experienced radiologists and was not based on random selection. Furthermore, each group was subdivided based on specific radiological features: Group 1 included cases with either definite dehiscence or suspected dehiscence, while Group 2 included cases showing either a thin bone roof over the canal or complete absence of dehiscence.

### 2.4. Statistical Analysis

Statistical analyses of the data were performed using SPSS version 23.0 software (IBM SPSS Inc., Chicago, IL, USA). CT was considered the reference method to evaluate the diagnostic performance of MRI for superior and posterior SCs. The agreement between CT and MRI results was evaluated using Cohen’s kappa statistics. CT-MRI agreement was determined by calculating sensitivity, specificity, positive predictive value (PPV), and negative predictive value (NPV). The kappa coefficient was analyzed for significance, and 95% confidence intervals were calculated for sensitivity and specificity values. Intergroup comparisons were performed using the Chi-square test or Fisher’s exact test. All statistical tests were performed as two-tailed and the significance level was accepted as *p* < 0.05. Missing data were excluded from the study, and subgroup analyses were performed separately according to the findings obtained in the Poschl and Stenver planes. Kappa values were interpreted as follows: 0.00–0.20 as poor, 0.21–0.40 as fair, 0.41–0.60 as moderate, 0.61–0.80 as substantial, and 0.81–1.00 as almost perfect agreement. For the analysis of CT and MRI agreement in semicircular canal evaluation, the results were stratified by CT finding categories (normal, definite dehiscence, suspected dehiscence, thin bone roof) and the distribution of MRI findings within each category was systematically analyzed across different MRI planes.

## 3. Results

A total of 103 patients who underwent CT examination of the temporal region were included in our study. The clinical indications for CT examination and the corresponding imaging findings were classified and expressed as percentages to reflect their distribution within the study population of 103 patients. When the reasons for CT examination of the patient group are analyzed, it is seen that the most common reason is hearing loss, with a rate of 48.5%. This was followed by chronic otitis media (COM), with 45.6%. Other common reasons included tinnitus (18.4%), vertigo and loss of balance (14.0%) and otalgia (14.6%). The most common finding on CT scans was COM (40%), followed by mastoid inflammation (20.4%). Normal CT findings were found in 14% of the patients (Table 1). Including these data provides essential context for understanding the clinical spectrum of semicircular canal (SC) pathology encountered in routine imaging practice. It also demonstrates the frequency and diagnostic yield of CT in detecting dehiscence or suspicious SC findings among patients referred for various otologic symptoms. This broad overview supports the relevance and necessity of subsequent MRI evaluation in a subset of patients, thereby reinforcing the rationale for this study’s prospective component.

A total of 206 semicircular canals were evaluated across all CT examinations, including 106 superior semicircular canals (SSCs) and 100 posterior semicircular canals (PSCs). Among the SSCs, 43 canals were identified as having or being suspected of having dehiscence—21 were classified as definite dehiscence, and 22 as suspected. A thin bony roof was detected in 19 of 163 normal superior semicircular ducts. In the posterior SCs, a total of 19 canals were found to have or to be suspected of having dehiscence, 7 of which were reported as definite dehiscence and 12 of which were reported as suspected dehiscence. A thin bone roof was present in 13 of 187 posterior SCs which were considered normal (Table 2).

When the MRI and CT results of the superior SCs were compared, 17 of the 20 canals evaluated as normal on CT were also evaluated as normal in the MRI final decision. In 9 of the 11 canals in which a definite dehiscence was found on CT, the presence of a definite dehiscence was confirmed in the MRI final decision. In 5 of the 11 canals with suspected dehiscence on CT, suspected dehiscence was also detected on the MRI final decision and definite dehiscence was seen in three of them. In 7 of 11 canals in which thin bone roof was detected, the same finding was reported in the MRI final decision. When the MRI planes were evaluated separately, it was observed that Stenver and Poschl planes, which were obtained during the examination for the first time in the literature, gave different results compared to the axial and coronal planes in the detection of dehiscence (Table 3).

When the MRI and CT results of the posterior SCs were compared, 33 of the 35 canals that were evaluated as normal on CT were also evaluated as normal on MRI. In 2 of the 3 canals in which definite dehiscence was found on CT, the presence of definite dehiscence was confirmed in the MRI final decision. In 4 of the 8 canals with suspected dehiscence on CT, definite dehiscence was detected on the MRI final decision and suspected dehiscence was found in one canal. In 5 of the 8 canals with thin bone roof, the same finding was reported in the MRI final decision (Table 4).

In the diagnostic performance evaluation, the sensitivity and specificity of the MRI final decision for the detection of SSCD were calculated as 78% and 92%, respectively, with reference to CT. In the detection of PSCD, the sensitivity and specificity of the MRI final decision were found to be 70% and 97%, respectively. When MRI planes were evaluated separately, the highest sensitivity for superior semicircular canal was found in axial and coronal planes (68%) and the highest specificity was found in axial plane (93%). For the posterior semicircular canal, the highest sensitivity was measured in the coronal plane (70%) and the highest specificity in the axial plane (95%).

The agreement analysis showed that the MRI final decision matched the CT results in detecting SSCD (κ = 0.71, *p* < 0.001) and PSCD (κ = 0.73, *p* < 0.001). The diagnostic consistency between the MRI and CT findings reached a high level of agreement. When agreement between different MRI planes and CT were analyzed for SSC detection, the axial (κ = 0.63, *p* < 0.001) plane showed substantial agreement, while the Poschl (κ = 0.43, *p* = 0.002) and Stenver (κ = 0.51, *p* < 0.001) planes showed moderate agreement with CT. The coronal plane (κ = 0.51, *p* = 0.118) did not show a statistically significant agreement. For PSC detection, all MRI planes showed significant agreement with CT, with kappa values ranging from 0.45 to 0.60, indicating moderate to substantial agreement across imaging orientations (Table 5).

In the case example, a 19-year-old female patient presented with complaints of decreased hearing on the right ear and a temporal CT scan revealed soft tissue densities in the right ear in favor of COM and dehiscence in the left SSC. On MRI, in all sequences (axial, coronal, Stenver, Poschl), dehiscence was easily visible and dura–perilymph contact was confirmed. In this case, the MRI findings were fully compatible with those of CT (Figure 3).

Using standardized evaluation criteria, interobserver agreement analysis showed substantial consistency in both CT and MRI assessments, with kappa values of 0.76 (SSC) and 0.73 (PSC) for CT, and average values of 0.77 (SSC) and 0.71 (PSC) for MRI. Agreement levels varied across MRI planes, with the highest in the Poschl plane for SSC (κ = 0.83) and the lowest in the axial plane for PSC (κ = 0.65) (Figure 4). These findings demonstrate overall high interobserver reliability, particularly in the evaluation of SSC on the Poschl plane.

## 4. Discussion

This research investigates MRI’s effectiveness for imaging diagnosis of semicircular canal dehiscence. The gold-standard status of CT exists but MRI has not received sufficient evaluation for its diagnostic capabilities. Our research examined MRI’s effectiveness in diagnosing superior and posterior SCD while assessing the new Poschl–Stenver plans from MRI examinations for the first time in the medical literature. MRI demonstrates value as a diagnostic tool for SCD evaluation because it provides satisfactory sensitivity and specificity results while avoiding ionizing radiation.

In our study, the sensitivity and specificity of MRI were found to be 78% and 92%, respectively, in the detection of SSCD with reference to CT. For PSC, the sensitivity and specificity of MRI were found to be 70% and 97%, respectively. These values indicate that MRI may be an important diagnostic method in the diagnosis of SCs. The high sensitivity and specificity values we obtained support that, although CT is still the gold standard, MRI may be a reliable alternative, especially in patient groups where radiation exposure should be avoided. Inal et al. [13] reported that 5.9% of SSC dehiscence was detected by MRI, 5.5% by multi-detector computed tomography (MDCT), and 5.1% was diagnosed by both methods. The results match our MRI-CT agreement values of κ = 0.71 (*p* < 0.001) for SSC and κ = 0.73 (*p* < 0.001) for PSC, which shows substantial agreement between the two imaging modalities. Eberhard et al. [14] reported that MRI plays a complementary role to CT in the diagnosis of SCD, but may give 20–39% false positive results in the presence of thin bone. This emphasizes the importance of the high specificity values in our study. In our study, the inclusion of images obtained, especially in the Poschl and Stenver plans, in the evaluation probably decreased the false positive rate and increased the specificity. Ionescu [15] et al. emphasized that fusion imaging of 3T MRI with high-resolution CT shows the relationship between SSC and PSC more sensitively, which supports our findings of high sensitivity and specificity. Although our study was performed with 1.5T MRI, the values we obtained are close to the results of 3T fusion imaging. This shows the contribution of additional sequences obtained in the Poschl and Stenver plans to the diagnosis. The superior semicircular canal dehiscence–semicircular canal imaging analysis tool (SSCD-SIAT) diagnostic tool developed by Fritz et al. [16] predicts the presence of SSCD with a 70% probability when a score of ≥6 is obtained (area under the curve (AUC) = 0.814), which is similar to our diagnostic performance with MRI. When clinical scoring systems such as SSCD-SIAT are combined with MRI findings, it can be considered that the diagnostic accuracy may increase even more.

In our study, taking additional sequences in the Poschl and Stenver plans prolonged the examination time. These sequences added ~5 min per scan. While this is longer than post-processing reformatting, the diagnostic value and radiation-free advantage justify the time. The effectiveness and agreement of different MRI planes with CT in the diagnosis of semicircular canal dehiscence was analyzed in detail in our study. For SSC, sensitivity was found to be 68% and specificity 93% in the axial plane, while sensitivity was found to be 60% and specificity 95% in the axial plane for PSC. Gurbuz et al. [17] emphasized that the use of Poschl and Stenver planes increased the diagnostic specificity in the evaluation of SSC bone roof thickness. Similarly, Öztürk et al. [9] stated that focal bone discontinuity should be observed in both perpendicular planes (Poschl and Stenver) reformatted in order to identify dehiscence. These findings support the diagnostic contribution of Poschl and Stenver planes acquired directly during the examination without the need for reformatting. Moreover, their implementation is feasible in routine clinical settings and does not significantly compromise patient comfort.

In our study, 20 SSCs that were found to be normal on CT were found to be normal in different MRI planes at varying rates (axial: 17, coronal: 16, Poschl: 18, Stenver: 14). Matic et al. [18] reported that MRI was evaluated in axial, parallel, perpendicular and oblique sagittal planes for the diagnosis of SSCD and that different planes had diagnostic value. In this study, it was emphasized that the MRI and CT results were completely compatible in 14 cases. In our study, the highest normal detection in the Poschl plane (18/20) indicates that this plane is particularly valuable in the evaluation of SSC. Connor et al. [19] showed that there was no significant difference between the CT and MRI measurements (*p* > 0.05) and reported 93% agreement in the diagnosis of large vestibular aqueduct syndrome/large endolymphatic sac anomaly (LVAS/LESA). This finding is in parallel with the 93% specificity value we found for SSC in the axial plane. In the same study, it was reported that midpoint measurements showed 95% agreement between CT and MRI, while operculum measurements were less consistent with 90% agreement. These results support that MRI can show a high degree of agreement with CT even in precise anatomical measurements. Johanis et al. [20] reported that high-resolution temporal bone CT scans were used for the diagnosis of SSCD and MRI was preferred for the diagnosis of endolymphatic hydrops (EH). This helps us to understand the variability in the diagnostic value of different planes. Nada et al. [21] emphasized that high-resolution CT is used as the main modality in the evaluation of temporal bone anomalies, but MRI is superior to CT in the evaluation of sensorineural hearing loss in terms of better visualization of the membranous labyrinth and cochlear nerve.

The Poschl and Stenver plans derived from primary MRI planning added extra diagnostic details beyond standard axial and coronal imaging results. The sequences acquired in these planes become essential for evaluating bone structures, particularly when dehiscence is suspected. The evaluation of all planes together resulted in the highest diagnostic accuracy through the ‘MRI final decision’.

This research examined posterior semicircular canal (PSC) dehiscence through extensive analysis of its characteristics. The analysis showed that definite dehiscence or suspected dehiscence occurred in 19 PSC canals, with 7 confirmed definite dehiscence and 12 suspected dehiscence. The results obtained from this study contribute essential information to posterior semicircular canal dehiscence diagnosis when compared to other published research. The radiological detection of posterior semicircular canal dehiscence revealed four total cases according to Whyte et al. [22], with two right, two left and one bilateral occurrence. The number of cases detected in our study exceeds the findings of other research, which supports that PSC dehiscence occurs infrequently. The research by Castellucci et al. [2] used high-resolution temporal bone CT to verify PSCD in six detailed patient examinations. Our study population characteristics together with standardized evaluation criteria likely contributed to the increased number of cases observed in this research.

In our study, the specificity of MRI in PSC was found to be 97%, which is a very high diagnostic value. Chemtob et al. [23] emphasized that postoperative high-resolution T2-weighted MRI is effective in determining the extent of canal occlusion by detecting the fluid space in the SCs. Fluid void was observed in all 13 cases, supporting the reliability of MRI in canal assessment. The primary images obtained without reformatting in Poschl and Stenver planes were applied during the examination for the first time in the literature, especially in the evaluation of PSC, and it is thought that this approach contributed to the high specificity value. In 4 out of 8 PSCs with suspected dehiscence on CT, definite dehiscence was detected by MRI, indicating that the combination of imaging modalities increases diagnostic accuracy. Waldeck et al. [10] reported that high-resolution CT offered high sensitivity in the diagnosis of dehiscence and provided a positive predictive value of 93% at a slice thickness of 0.5 mm. Similarly, Chemtob et al. [23] reported that when bone defects detected by preoperative CT were compared with postoperative MRI, residual defects were confirmed in some cases. These findings support the potential of MRI to clarify suspicious cases. The high specificity value obtained in our study (97%) indicates that MRI is highly reliable, especially in the exclusion of PSCD. This has clinical importance in terms of preventing unnecessary further investigations and reducing radiation exposure. Poschl and Stenver planes, which were obtained by planning during the examination for the first time in the literature, showed substantial agreement with CT and proved to have diagnostic value.

In our study, the presence of a thin bone roof in the SCs was an important finding. A thin bone roof was found in 19 of the superior SCs and 13 of the posterior SCs. Similarly, Davvaz et al. [24] reported a ‘papyraceous’ pattern, which they defined as a thin bone roof in the SSC, in 34 cases (6.1%) and detected a dehiscence pattern in 43 cases (7.7%) in the SSC 22. These findings support that the presence of thin bone roof is an important finding in radiological evaluations. In 5 of 11 cases of SSC with suspected dehiscence on CT, suspected and definite dehiscence was detected on MRI in five and three cases, respectively. Waldeck et al. [9] reported that high-resolution CT with a slice thickness of 0.6 mm had a positive predictive value of 93% for the correct diagnosis of SSCD and the diagnosis was confirmed by multiplanar reconstructions. They also emphasized that a thin section thickness of 0.6 mm increased sensitivity in the diagnosis of dehiscence, whereas thicker sections increased the risk of misdiagnosis [10]. In our study, CT with a slice thickness of 1 mm was used, and it is thought that the sensitivity values may increase if thinner slices are taken.

In our study, in 7 of 11 SSC cases with a thin bone roof on CT, the same finding was detected in the MRI final decision. However, the results varied according to MRI planes. In our study, the coronal plane did not show a significant agreement with CT for SSC detection (κ = 0.51, *p* = 0.118), unlike the other MRI planes. This may be explained by the fact that the coronal plane is not optimally aligned with the anatomical orientation of the superior semicircular canal, which is more accurately visualized in oblique reformatted planes such as Poschl or Stenver. Kraus et al. [25] reported that CT and MRI findings for the diagnosis of central vestibular disorder are not always consistent, with minimal changes in some cases. This is similar to our variable results obtained in different MRI planes. Especially Poschl and Stenver planes, which we obtained during the examination for the first time in the literature, were observed to provide different sensitivity/specificity profiles in the evaluation of thin bone structures. In the study by Horatiu et al., erosion or marked thinning of the tegmen tympani were detected on high-resolution CT in 13 cases but confirmed intraoperatively in nine cases. The sensitivity and specificity of high-resolution CT in detecting this finding was 90% and 88.8%, respectively [26]. This is in parallel with our results showing concordance between CT and MRI. In the same study, the reliability of high-resolution CT in the evaluation of thin bone structures was emphasized, but it was also stated that it may be misleading in some cases. It is thought that MRI, especially Poschl and Stenver planes obtained by planning during the examination, may provide complementary information in such misleading situations. Rodrigues et al. [27] reported that dehiscence of the semicircular canals is seen in approximately 9% of patients with Down syndrome. They also reported that 75% of inner ear malformations can be easily evaluated by CT and/or MRI. This shows the capacity of imaging modalities to detect such anomalies, in agreement with our results, which varied according to MRI planes.

The evaluation of thin bone roof serves as a crucial method to detect early semicircular canal dehiscence and risky anatomical variants. Our research showed that MRI produced results which matched the CT findings when using standardized evaluation criteria for thin bone roof detection. The results indicate that MRI provides useful information for both dehiscence detection and the assessment of thin bone roof structures.

Our research showed significant findings when we analyzed radiological data against clinical signs and symptoms of SCD. The main patient complaints consisted of hearing loss at 48.5%, followed by chronic otitis media (COM) at 45.6% and tinnitus at 18.4%. Berning et al. [28] found that symptomatic patients mainly sought CT scans because of hearing loss (20.1%), vertigo (5.5%), and tinnitus (5.5%). Cocca et al. [29] observed that hearing loss together with tinnitus and loss of balance appeared as clinical symptoms, which correlated with the imaging results in a patient with SCD. Our research results match the existing literature but hearing loss occurred more frequently among our study participants.

The research revealed that 14% of patients experienced vertigo and loss of balance. Romiyo et al. [30] observed vertigo in 89% of SSCD patients and balance loss in 92.5% of patients before surgery. The lower rate in our study may be due to the fact that our study population was not specific for SCD and included patients who underwent CT scan for different indications. Kurihara et al. [31] noted that vertigo and balance disorders occur frequently among patients who have otic capsule injuries. The different clinical presentations of SCD become apparent through this comparison, which demonstrates the need for imaging modalities in diagnosis.

Normal CT findings were found in 14% of the patients in our study. Aladham et al. [32] reported that middle ear structures, ossicular chain and mastoid cells were normal in a patient diagnosed with SSCD, and only SSCD was detected as a specific finding. Campion et al. [33] emphasized that, in some cases, especially in early-stage inflammatory processes, CT findings may appear normal. This shows that radiological findings may not always be evident even in symptomatic patients, highlighting the importance of considering further diagnostic tools, such as MRI, especially when initial imaging is inconclusive.

The Poschl and Stenver plans used for MRI imaging during the examination provided substantial radiological agreement with clinical signs and symptoms for the first time in the medical literature. The standardized evaluation showed that MRI provides diagnostic value when clinicians suspect SCD, especially when the CT results appear normal. This diagnostic method provides an essential non-radiation-based solution to verify or rule out SCD in patients who show comparable symptoms.

The limitations of our study include possible biases arising from the retrospective and prospective design, limited sample size, and generalizability problems of a single-center experience. In addition, the MRI examinations were conducted using a 1.5T scanner, which may offer lower spatial resolution compared to more advanced 3T systems, potentially limiting the detection of small or borderline dehiscences. Furthermore, the CT scans were performed with a slice thickness of 1 mm, which might have reduced sensitivity in identifying subtle bony defects. Future studies should consider using thinner CT slices (≤0.6 mm) to minimize false-negative results. To validate and strengthen our findings, future studies employing higher-resolution imaging protocols are recommended. Our strengths are the use of Poschl and Stenver planes planned during the examination for the first time in the literature, a detailed comparison of different imaging planes, standardized evaluation criteria, and blinded double reading.

The Poschl and Stenver planes showed lower agreement value than axial slices in the diagnosis of SCD and we encountered some difficulties when we wanted to use these two specialized planes: (i) it is difficult for MRI technicians to find the SSC and PSC axes on axial images to make an acquisition plan; (ii) with the parameters we used, MRI acquisitions in Poschl and Stenver planes could be completed in approximately 5 min each, which is significantly longer than the time it takes to create Poschl and Stenver reformats in the post-processing stage of CT or MR examination. These practical difficulties may limit the introduction of the method into clinical routine practice.

In future studies, multicenter prospective studies with larger patient groups, agreement between intraoperative findings and radiological images, and long-term clinical follow-up will help to determine the optimum imaging protocols in the diagnosis of semicircular canal dehiscence.

## 5. Conclusions

In conclusion, MRI demonstrated similar diagnostic precision to CT when evaluating posterior and superior SCD. The MRI diagnostic accuracy for SSCD reached 78% sensitivity and 92% specificity, and for PSCD, it reached 70% sensitivity and 97% specificity. The diagnostic accuracy values suggest that MRI may serve as a viable alternative to CT in selected cases for the evaluation of semicircular canal dehiscence but should not be considered a full replacement.

The images from the Poschl and Stenver planes were acquired for the first time in the literature by planning during the examination instead of post-processing reformatting, which increases the diagnostic accuracy. The sequences obtained in these planes showed a moderate agreement with CT, but a substantial agreement was found between the ‘MRI final decision’, in which all MRI planes were evaluated together, and CT (κ = 0.71, *p* < 0.001 for the superior canal; κ = 0.73, *p* < 0.001 for the posterior canal).

Therefore, it would be useful to pay attention to the roofs of the SCs in MRI examinations performed for complaints such as vertigo, nystagmus, and hearing loss, particularly in the absence of other findings to explain these symptoms, and to report suspicious cases accordingly. In patients with normal MRI findings, this approach may avoid the need for additional CT examinations. Cases found to be suspicious on MRI or CT scans with high collimations, such as 1 mm, should be examined with CT scanners with fine collimation in order to move towards a definitive diagnosis.

Clinical practice benefits from MRI because this imaging technique uses no ionizing radiation and produces consistent results between different observers. The Poschl and Stenver sequences represent a valuable diagnostic tool because they represent the first time these sequences have been used in the literature for primary planning during examinations. These sequences should be included in standard MRI protocols for SCD evaluation.

## Figures and Tables

**Figure 1 brainsci-15-00555-f001:**
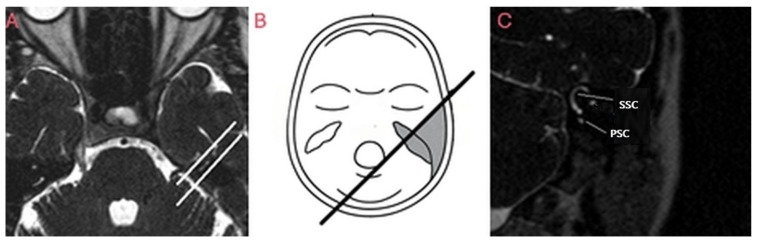
(**A**–**C**): MRI acquisition in Poschl plane (from our case). (**A**,**B**) Determination of the plane perpendicular to the long axis of the posterior semicircular canal (PSC) on axial images. (**C**) Superior semicircular canal (SSC) is visualized along its long axis, while posterior semicircular canal (PSC) is seen in orthogonal plane.

**Figure 2 brainsci-15-00555-f002:**
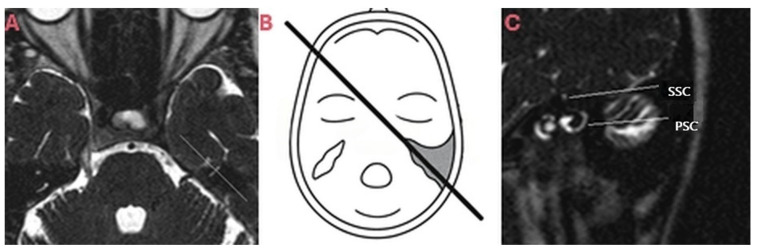
(**A**–**C**)**:** MRI acquisition in Stenver plane (from our case). (**A**,**B**) Determination of the plane perpendicular to the long axis of the superior semicircular canal (SSC) on axial images. (**C**) Posterior semicircular canal (PSC) is visualized along its long axis, while superior semicircular canal (SSC) is seen in orthogonal plane.

**Figure 3 brainsci-15-00555-f003:**
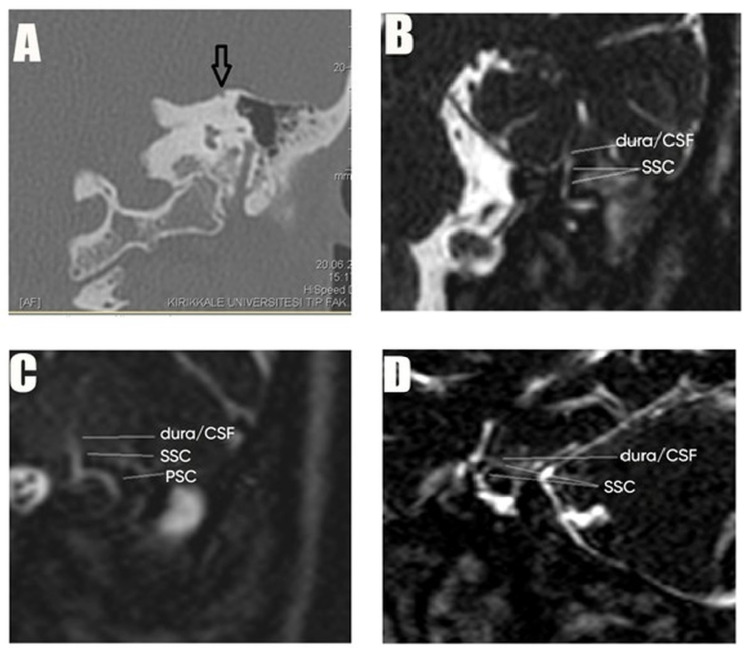
(**A**–**D**): Imaging of inner ear abnormalities using CT and MRI scans. (**A**) CT coronal: a bone defect is visible in the superior semicircular canal (SSC), indicated by an arrow. (**B**) MRI coronal: the dura/cerebrospinal fluid (CSF) interface and the SSC are visible, showing their anatomical relationship. (**C**) MRI Stenver: dural contact with the SSC is observed orthogonally, highlighting the proximity of the dura to the canal. (**D**) MRI Poschl: dural contact with the SSC is seen along its long axis, providing a detailed view of the interaction between the dura and the canal.

**Figure 4 brainsci-15-00555-f004:**
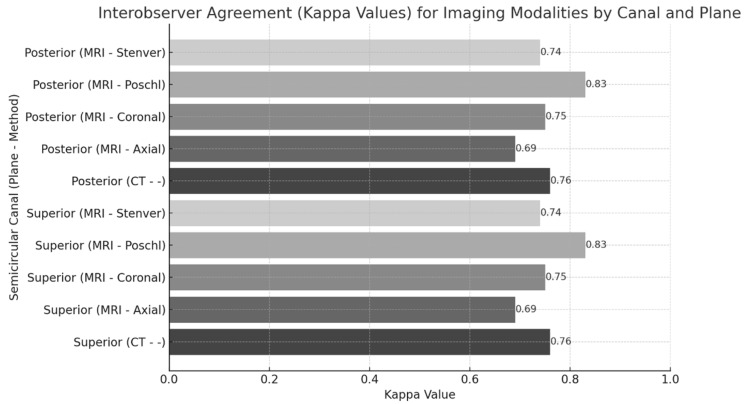
Interobserver agreement (kappa values) for imaging modalities by canal and plane: bar graph showing kappa values for superior semicircular canal (SSC) and posterior semicircular canal (PSC) across different imaging modalities (CT and MRI) and planes. CT: computed tomography; MRI: magnetic resonance imaging.

**Table 1 brainsci-15-00555-t001:** Distribution of CT request reasons and CT findings.

CT Request Reasons	*n* = Patients (%)	CT Findings	*n* = Patients (%)
Vertigo—balance loss	15 (14.0%)	Normal	29 (14.0%)
Hearing loss	50 (48.5%)	COM	42 (40.0%)
Tinnitus	19 (18.4%)	Mastoid inflammation	21 (20.4%)
COM	47 (45.6%)	Ossicular chain defect	12 (11.7%)
Otalgia	15 (14.6%)	Surgical defect	11 (10.7%)
Hyperacusis	3 (2.9%)	High jugular bulb	5 (4.8%)
Tympanic membrane perforation	12 (11.7%)	Ossicular chain sclerosis	4 (3.9%)
Ear fullness	2 (1.9%)	Cholesteatoma	3 (2.9%)
Cranial dysmorphism	1 (1.0%)	Cranial dysmorphism	1 (1.0%)
Eagle syndrome	2 (1.9%)	Eagle syndrome	2 (1.9%)
Trauma	3 (2.9%)	Skull base fracture	2 (1.9%)

CT: computed tomography; COM: chronic otitis media. Percentages are calculated based on the total number of patients (*n* = 103 for request reasons, *n* = 103 for findings).

**Table 2 brainsci-15-00555-t002:** Results of all CT examinations included in the study, including findings for the SSC and PSC.

Findings	SSC	PSC
Group 1: Dehiscence/suspected dehiscence	43	19
-Definite dehiscence	21	7
-Suspected dehiscence	22	12
Group 2: No dehiscence	163	187
-Thin bone roof	19	13
-Normal	144	174
Total	206	206

SSC: superior semicircular canal; PSC: posterior semicircular canal. Group 1 includes cases with definite or suspected dehiscence, while Group 2 includes normal or thin bone roof cases.

**Table 3 brainsci-15-00555-t003:** Distribution of MRI findings across different planes compared to CT findings for superior semicircular canal (SSC) evaluations.

CT Findings	MRI Results	MRI Axial (*n* = 53)	MRI Coronal (*n* = 53)	MRI Poschl (*n* = 53)	MRI Stenver (*n* = 53)	MRI Final Decision (*n* = 53)
**Normal** **(*n* = 20)**	Total	20	20	20	20	20
	Normal	17	16	18	14	17
	Definite dehiscence	0	0	1	1	0
	Suspected dehiscence	1	2	0	1	1
	Thin bone roof	2	2	1	4	2
**Definite** **dehiscence** **(*n* = 11)**	Total	11	11	11	11	11
	Normal	1	1	2	1	1
	Definite dehiscence	3	7	7	5	9
	Suspected dehiscence	5	3	2	3	1
	Thin bone roof	2	0	0	2	0
**Suspected** **dehiscence** **(*n* = 11)**	Total	11	11	11	11	11
	[MRI Results’ Distribution...]					
**Thin bone** **roof** **(*n* = 11)**	Total	11	11	11	11	11
	[MRI Results’ Distribution...]					

Table note: CT: computed tomography; MRI: magnetic resonance imaging. Each row shows the distribution of MRI findings for each CT finding category across different MRI planes. “MRI Final Decision” represents the consensus evaluation across all MRI planes. Classification: Normal = absence of any abnormality; Definite dehiscence = complete absence of bone roof; Suspected dehiscence = incomplete or uncertain bone discontinuity; Thin bone roof = intact but attenuated bone coverage (≤0.5 mm).

**Table 4 brainsci-15-00555-t004:** Distribution of MRI findings across different planes compared to CT findings for posterior semicircular canal (PSC) evaluations.

CT Findings	MRI Results	MRI Axial (*n* = 54)	MRI Coronal (*n* = 54)	MRI Poschl (*n* = 54)	MRI Stenver (*n* = 54)	MRI Final Decision (*n* = 54)
**Normal** **(*n* = 35)**	Total	35	35	35	35	35
	Normal	33	31	31	33	33
	Definite dehiscence	0	2	2	0	0
	Suspected dehiscence	0	0	1	0	0
	Thin bone roof	2	2	1	2	2
**Definite** **dehiscence** **(*n* = 3)**	Total	3	3	3	3	3
	Normal	1	1	1	1	1
	Definite dehiscence	1	1	2	2	2
	Suspected dehiscence	1	1	0	0	0
	Thin bone roof	0	0	0	0	0
**Suspected** **dehiscence** **(*n* = 8)**	Total	8	8	8	8	8
	Normal	2	3	2	2	2
	Definite dehiscence	3	3	4	4	4
	Suspected dehiscence	1	2	0	1	1
	Thin bone roof	2	0	2	1	1
**Thin bone** **roof** **(*n* = 8)**	Total	8	8	8	8	8
	Normal	1	4	2	2	2
	Definite dehiscence	2	3	1	1	1
	Suspected dehiscence	0	1	0	0	0
	Thin bone roof	5	0	5	5	5

Table note: CT: computed tomography; MRI: magnetic resonance imaging. This table shows how CT findings for PSC were interpreted on different MRI planes. Each row represents the distribution of MRI results for a specific CT finding category. The “MRI Final Decision” represents the consensus evaluation across all MRI planes. Classification criteria: Normal = absence of any abnormality; Definite dehiscence = complete absence of bone roof; Suspected dehiscence = incomplete or uncertain bone discontinuity; Thin bone roof = intact but attenuated bone coverage (≤0.5 mm).

**Table 5 brainsci-15-00555-t005:** Diagnostic performance of MRI for SSCD and PSCD: sensitivity, specificity, PPV, NPV, and agreement results across MRI planes compared to CT (reference standard).

Canal	MRI Planes	Sensitivity (%) (95% CI)	Specificity (%) (95% CI)	PPV (%)	NPV (%)	Agreement (κ)	*p*-Value
**SSCD**	Axial	68 (49.7–86.3)	93 (82.9–100.0)	88	80	0.63 *	<0.001
	Coronal	68 (49.7–86.3)	83 (69.6–96.4)	75	78	0.51	0.118
	Poschl	63 (44.1–81.9)	80 (65.7–94.3)	70	75	0.43 *	0.002
	Stenver	63 (44.1–81.9)	87 (75.1–98.9)	77	77	0.51 *	<0.001
	MRI Final Decision	78 (62.0–94.0)	92 (81.4–100.0)	83	77	0.71 *	<0.001
**PSCD**	Axial	60 (35.2–84.8)	95 (88.3–100.0)	75	91	0.60 *	<0.001
	Coronal	70 (47.1–92.9)	86 (75.7–96.3)	53	92	0.48 *	<0.001
	Poschl	60 (35.2–84.8)	88 (78.5–97.5)	54	90	0.45 *	<0.001
	Stenver	60 (35.2–84.8)	88 (78.5–97.5)	54	90	0.46 *	<0.001
	MRI Final Decision	70 (47.1–92.9)	97 (91.6–100.0)	87	93	0.73 *	<0.001

Table note: CT: computed tomography; MRI: magnetic resonance imaging; PPV: positive predictive value; NPV: negative predictive value; SSCD: superior semicircular canal dehiscence; PSCD: posterior semicircular canal dehiscence; κ: Cohen’s kappa coefficient. Asterisk (*) indicates statistically significant agreement (*p* < 0.05). Kappa values: 0.00–0.20 = poor, 0.21–0.40 = fair, 0.41–0.60 = moderate, 0.61–0.80 = substantial, 0.81–1.00 = almost perfect agreement. MRI Final Decision represents consensus across all planes.

## Data Availability

The data presented in this study are available on request from the corresponding author due to privacy and ethical restrictions.

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
