# Peer review of "Is CT Still the Gold Standard in Semicircular Canal Dehiscence? Diagnostic Value of MRI in Poschl and Stenver Planes"

_brainsci, 2025, doi:10.3390/brainsci15060555_

Round 1

Reviewer 1 Report

Comments and Suggestions for Authors

This study investigates whether MRI in Poschl and Stenver planes can reliably diagnose semicircular canal dehiscence (SCD) compared to CT. The topic is clinically relevant and well-motivated, but the current manuscript requires major revision before being considered for publication. The main concerns relate to the study design, clarity of presentation, and interpretation of results

  1. In Table 1, how did the % values of each item calculated?
  2. It is unclear how the 27 patients were selected for MRI from the original 103. Please explain inclusion criteria, study timeline, and whether patient selection introduces potential bias.
  3. The definitions of “definite dehiscence”, “suspected dehiscence”, and “thin roof” are not clearly described. Provide specific diagnostic thresholds or criteria used for classification on CT and MRI.
  4. Using Pearson correlation to assess agreement between CT and MRI is inappropriate for categorical diagnostic data. Consider using kappa statistics or cross-tabulations. Also, p-values should be reported as p < 0.001, not p = 0.000. Include confidence intervals for sensitivity/specificity.
  5. The conclusion that MRI could replace CT seems too strong given the reported sensitivities (70–78%). Please revise the language to reflect that MRI may serve as an alternative in selected cases, but not a full replacement.

Reviewer 2 Report

Comments and Suggestions for Authors

This manuscript presents a comparative evaluation of CT and MRI in diagnosing semicircular canal dehiscence (SCD), emphasizing the diagnostic value of MRI sequences obtained in the Poschl and Stenver planes. It stands out as the first study in the literature to acquire these sequences prospectively during MRI examination rather than through post-processing, which is a meaningful contribution to diagnostic radiology and otology.
The research addresses a timely and clinically relevant question, especially in minimizing radiation exposure. It is methodologically sound, with a balanced retrospective-prospective design, and employs rigorous image evaluation criteria and interobserver agreement assessment. However, the paper would benefit from better organization, tighter phrasing, reduced repetition, clearer figure/table integration, and a more focused discussion of clinical implications.
Bellow is some suggestions to improve the manuscript:

 Page 1

  • Line 35–36 "magnetic resonance imaging (MRI) of the superior and posterior semicircular canals (SCs) with and without dehiscence on computed tomography (CT)”,  rephrasing is needed>.-“…MRI of the superior and posterior semicircular canals (SCs) in cases with and without dehiscence observed on CT...”
  • Line 38 "resonance" was split incorrectly in the line above. Suggest fixing hyphenation: "magnetic reso-nance" -> "magnetic resonance"
  • Line 15  "post process reformatting" -> "post-processing reformatting"
  • Line 27 "do not need additional CT scans to rule out SCD" ->could benefit from hedging language, e.g., “may not require...”
  • Line 31–32, Keywords are inconsistent in punctuation (some lowercase, some upper), and “posterior semisircular canal” has a typo. “semisircular” -> “semicircular”

Page 2

  • Line 41 "balance disturbance" ->more commonly “balance disorders”
  • Line 44 "radiological diagnosis" ->consider “imaging diagnosis” for precision
  • Line 52 Important limitation: CT may give false positives. Consider citing more explicitly how often this occurs or under what imaging conditions.

Page 3–4

  • Line 77 "appearance or suspicion... thinner than normal”, grammatically clumsy -> “...with signs or suspicion of SCD, or with a canal roof thinner than normal...”
  • Line 81 “this SSC was excluded from the study”, unclear why only one canal and not the whole patient. Consider rephrasing for clarity.

Page 5–6

  • Line 143 "Missing or missing data" ->Typo/duplication. Use either "missing data" or "incomplete data."
  • Line 156 The number of SSCs and PSCs evaluated should be clearly restated here.
  • Table 2Good summary, but abbreviation “SSC” should be defined in caption for standalone clarity.

Page 8

  • Line 205–207Technical values are good, but not contextualized. Suggest clarifying: “This level of correlation suggests moderate agreement.”
  • Line 211 "coronal plane... did not show significant correlation" – suggest clarifying why this might be the case.

Page 10

  • Line 232 "Interobserver agreement analysis”, okay, but repetition could be reduced. Merge with next sentence.
  • Line 236  "SSC channel" –> redundant. SSC = canal. Suggest “SSC” alone.

Page 11–12

  • Line 278–280  Long sentence. Split for clarity-> Suggested: “These sequences added ~5 minutes per scan. While this is longer than post-processed reformats, the diagnostic value and radiation-free advantage justify the time.”
  • Line 291 Strong data; would be strengthened by mentioning patient comfort or scan feasibility in routine settings.

Page 13–14

  • Line 370 “different diagnostic contributions” ->vague. Be more specific: “...different sensitivity/specificity profiles.”
  • Line 389–396 Patient symptom correlation is valuable. Consider organizing as a table or clearer bullet point summary to make links between symptom and imaging results more readable.

Page 15

  • Line 411 “radiological findings may not always be evident even in symptomatic patients” ->excellent point; strengthen by mentioning possible implications for clinicians (e.g., MRI as a second-line test).

Page 16–17

  • Line 447–450 Policy implication: “...no additional CT examination will be required...” – change to more cautious wording:-> Suggest: “...may avoid the need for additional CT in patients with normal MRI findings and relevant symptoms.”

Reviewer 3 Report

Comments and Suggestions for Authors

1-The study had a combined retrospective-prospective design, which introduced potential selection bias. ِ Authors must conduct a larger, multicenter prospective study to validate findings and enhance generalizability.

2-The sample size was limited (n=27 in the prospective cohort), reducing statistical power and generalizability.

3-MRI was performed using a 1.5T scanner, which may have lower resolution compared to 3T systems.

4-CT scans used a 1 mm slice thickness, potentially missing subtle dehiscences detectable with thinner slices. Authors should employ CT protocols with ≤0.6 mm slice thickness to minimize false negatives.

5-Practical challenges in acquiring Poschl/Stenver planes during MRI (e.g., technician expertise, prolonged scan time) were noted.

Reviewer 4 Report

Comments and Suggestions for Authors

Authors present a partial retrospective study on the use of MRI compared to CT in the study of Semicircular Canal Dehiscence. English in general is acceptable. The aims and introduction are clear, well-presented and well referenced. The methods could be improved with a protocol section and clarifying some questions I make in my comments. All tables in the Results sections should be checked and corrected/completed. They do not allow readers to grasp the ideas presented in this work or assess the scientific merit of it.  The discussion section is quite large except for the limitation section which is very brief.

Major comments

Last paragraph of the introduction says, “The primary aim of this study was to evaluate whether MRI gives results close to CT 63 by performing MRI on semicircular canals (SCs) with and without CT dehiscence.” Is it CT dehiscence or is it SCD or SSCD? It is confusing as it is.

Methodology is a bit unclear. The 103 can be ignored as study was performed on 27 patients which gives us the 53-54 auditory systems to study. What worries me is that CT imaging was performed in 2012 and MRI now in 2024 2025? Prospective studies author says. Am I right? Is this difference in timing not a major confounds of this study? Have subjects of study undergone surgery or any kind of treatment at this time? Don’t you need an ethics permit to perform all this new scanning? If I am wrong with all this, please make the study protocol clearer.

How was the division into groups performed? Radiologists? Scales? Please specify. Also, the use of dividing patients into two groups at the time is not clear to me.

Tables 1 and 2 are understandable. Do not really know why authors go back to the 103 CT patients. Thought this study would be centered around the 27 patients with MRI. Therefore, tables on this cohort, I believe does not provide much information towards the aims of the paper. Could be deleted.

Tables 3 and 4, the fist column of these tables, is very confusing. Extra explanation of the meaning of it should be provided. For a start, the first two rows of the first column present Normal (0), with different results in them. It makes no sense.

Table 5 needs also more explanation. Is it SSCD or SSC in the first column? Also. An indication of which images were obtained with CT and which with MRI is needed. All this to make sense of the results.

I understand results from figure 4 are the most relevant towards the aims of this study.

Discussion section is quite large with the exception of the limitation section which is very brief.

Minor comments

This sentence in abstract: “on computed tomography (CT) gives results similar to CT”, would it not make more sense without “on computed tomography (CT).

The second sentence of the abstract with the secondary objective should also be shorter and more specific.

In abstract number it is not clear the number of patients used for this study. Hopefully in the methods section this gets clearer.

Again, second objective in the last paragraph of the introduction could be presented in a clearer form.

Add reference in line 106

Round 2

Reviewer 1 Report

Comments and Suggestions for Authors

The current version can be accepted for publication.

Reviewer 4 Report

Comments and Suggestions for Authors

I have seen that in the very educated response from authors all my concerns have been addressed. I am happy with the responses from authors, and I believe the manuscript in this moment is much better than before revision.